# How climate policy commitments influence energy systems and the economies of US states

Parrish Bergquist [1] ✉ & Christopher Warshaw [2] ✉

In the United States, state governments have been the locus of action for addressing climate change. However, the lack of a holistic measure of state climate policy has prevented a comprehensive assessment of state policies' effectiveness. Here, we assemble information from 25 individual policies to develop an aggregate index of state climate policies from 2000-2020. The climate policy index highlights variation between states which is difficult to assess in single policy studies. Next, we examine the environmental and economic consequences of state climate policy. A standard-deviation increase in climate policy is associated with a 5% reduction in per-capita electricity-sector CO2 emissions and a 2% reduction in economy-wide CO2 emissions per capita. We do not find evidence that more stringent climate policy harms states' economies. Our results make clear the benefits of state climate policy, while showing that current state efforts are unlikelyto meet the US goal under the Paris Climate Accord.

Over the past two decades, US states have been leaders in enacting policies to mitigate climate change[1,2]. State governments have enacted and implemented a broad array of policies intended to reduce the greenhouse gas emissions that cause climate change. These include policies that restructure the market for electricity production and sale; standards requiring electricity producers to increase their use of non-fossil fuel sources; limits on pollution emissions from manufacturing facilities, cars, and trucks; and programs that incentivize businesses and individuals to install renewable energy technologies or use electricity more efficiently. States tend to adopt several of these policies in combination, leading to a wide array of state climate policy regimes.

Despite the diversity in the design, stringency, and comprehensiveness of states' climate policy regimes, scholars lack holistic assessments of the effectiveness of these efforts to reduce $CO_2$ emissions. Most of the extant literature assessing the impact of state climate policy focuses on renewable portfolio standards (RPS), which are one of the earliest-adopted and most widely used tools to promote a transition to a cleaner energy system[3–15]. In general, RPS policies have helped to increase renewable generation capacity in states that have implemented them[4,6,8,10,16], although we lack evidence that states with RPSs have a higher proportion of renewables in their energy mix than states without them[12]. Scholars have found mixed results concerning the independent impacts of other policies, such as public benefits funds (PBF), net metering (NEM), or green power options (GPO)[5,17–20].

Here we use Bayesian factor analysis[21,22] (see Methods for details) to estimate the stringency of states' climate policy regimes and gain analytical leverage on this concept. The benefit of the index is that, by pooling information across many measures, we produce a set of estimates that is more comprehensive than any single-policy measure would be. Our approach builds upon multi-disciplinary work using factor analysis and other dimension-reduction techniques to examine important concepts in governance[23–26], public opinion[27–32], individual and corporate behavior[33,34], and ecology[35,36] that are relevant to climate change. Our modeling approach weights different policies according to the information each provides about the state's overall regime. The approach also accounts for variation in different states' versions of the "same" policy instrument. The resulting index provides an overall, comparative ranking of state climate policy efforts.

We use our estimates of state climate policy to examine the environmental and economic consequences of state climate policies.

[1]Department of Political Science, University of Pennsylvania, Philadelphia, PA, USA. [2]Department of Political Science, George Washington University, Washington, DC, USA. ✉e-mail: pberg@upenn.edu; warshaw@gwu.edu

**Table 1 | Policies included in the dataset of state climate policy stringency**

| Policy | No. states enacting | State(s) adopting by year 1 |
|---|---|---|
| CA Car Emissions Standard | 18 | CA |
| Climate action plan | 33 | DE, HI, IL |
| Community Solar | 21 | MA |
| Complete Streets Policies | 32 | FL, OR, RI |
| Electric decoupling | 41 | AL, CT, MN, NH, VT |
| Emissions performance standards | 6 | OR |
| Energy efficiency resource standard (continuous) | 25 | TX, VT |
| Energy efficiency target | 36 | FL, TX, VT |
| Environmental Building Standards | 43 | MD, OR |
| Environmental Policy Act | 17 | CA, CT, DC, GA, HI, IN, MA, MD, MN, MT, NC, NJ, NY, SD, VA, WA, WI |
| Fuel generation mix disclosure | 25 | CA, CO, CT, DC, DE, FL, IL, MA, MD, ME, MI, NJ, NY, OH, OR, PA, VA, WA |
| GHG target | 27 | NH, RI, VA, VT |
| Gas decoupling | 38 | AL, CT, MN, NH, NV, VT |
| Gas tax | 51 | |
| Greenhouse Gas Cap | 14 | CT, DE, ME, NH, NJ, NY, VT |
| Greenhouse gas registry/ reporting | 42 | CA |
| Low-income energy efficiency programs | 44 | AK, AR, AZ, CA, CO, CT, DC, DE, FL, IA, ID, IL, IN, KS, KY, MA, MD, ME, MI, MN, MO, MS, MT, NC, NH, NJ, NM, NV, NY, OH, OK, OR, PA, RI, SC, TN, TX, UT, VA, VT, WA, WI |
| Mandatory green power option | 25 | IA, MN, WA |
| On-site renewable generation | 45 | CA, CT, DC, DE, IA, MD, ME, MN, MT, ND, NH, NJ, NV, NY, OH, OK, OR, VA, VT, WA, WI |
| PACE authorization | 37 | HI |
| Public Benefit Fund | 25 | CA, CT, DC, DE, IL, MA, ME, MI, MN, MT, NH, NJ, NY, OH, OR, PA, RI, VT, WI |
| RPS target (binding only) | 31 | AZ, IA, ME, WI |
| Renewable Portfolio Standard | 39 | AZ, CT, IA, MA, ME, NJ, NV, TX, WI |
| Solar Tax Credit | 40 | AZ, CA, CT, FL, HI, IA, IN, KS, LA, MA, MD, MN, MT, NC, NH, NJ, NV, NY, OR, RI, SD, TX, VA, VT, WI |
| State preemption of local gas bans | 4 | AZ, LA, OK, TN |

The table shows the name of each policy, the number of states that have adopted it, and the number of states that had adopted it by the first year for which we have a record of the policy's adoption. Our dataset includes policy data from all 50 states and the District of Columbia. Supplementary Table S3 provides a longer description of each policy.

We find that more stringent state climate policy regimes are associated with meaningful reductions in $CO_2$ emissions, and we do not find evidence that more stringent climate policies undermine economic growth in the states.

## Results

First, we aggregate information from 25 individual policies to develop a holistic index of state climate policies from 2000-2020. We focus on these years because this is the time period when states passed the bulk of their policies to address climate change. Next, we use the index to examine the environmental and economic consequences of state climate policy.

### Climate policy index

Previous efforts to assess climate policy have used coarse additive indices[37] or, more commonly, multivariate regressions with each policy included independently[5,17,20,38]. These approaches are limited, however, because additive indices do not appropriately account for variation in the targeted and realized impacts of different policy instruments. Additionally, various climate policies are often adopted in combination. As a result, their independent impacts can be difficult to disentangle using multivariate regression, due to multicollinearity. A holistic measure of variation over time in states' climate policy regimes would allow for more reliable assessments of the drivers and impacts of state efforts to reduce climate change[39].

Policy instruments vary in the extent and types of changes that industry and government actors will need to implement, in order to comply. States also enact different versions of the "same" policy instrument (eg, RPS or net metering). These design details are likely to influence policy impacts. In light of these particularities, some studies have sought to account for differences between states' RPS policy designs and stringency when assessing the effects of this policy[4,10,13,14,40,41]. Our climate policy index builds upon work that accounts for variation in RPS design in two ways. First, we examine climate policy regimes holistically by incorporating the many types of policies that states have enacted. Second, we account for variation in states' particular versions of the "same" policy instruments.

We compile a granular dataset that reflects adoption and design differences across states for each of 25 policies that multiple states have used to reduce greenhouse gas emissions, promote cleaner energy production, and boost energy efficiency (Table 1). The most widely adopted policies are those that incentivize on-site renewable electricity generation: net metering or feed-in tariffs. These allow solar energy system owners to get paid for electricity that their solar systems add to the grid. Many states have also adopted solar tax credits, renewable portfolio standards, electric and gas system decoupling, and requirements that power plants register and record their emissions. Fewer states have adopted greenhouse gas emission standards, greenhouse gas caps, and preemption of local natural gas bans (a policy intended to slow rather than accelerate the transition away from fossil fuels). Most of the policies in our dataset focus on the electricity sector. This reflects the reality that the electricity sector has received the bulk of states' climate policy effort, relative to other sectors.

We sought to include in our index all policies that could affect $CO_2$ emissions, can be systematically coded for all states over time, and are eligible for adoption in all states. In gathering and coding our data, we strove to strike a balance between breadth (ie, comprehensiveness over time and across states) and depth (ie, faithful depictions of finely

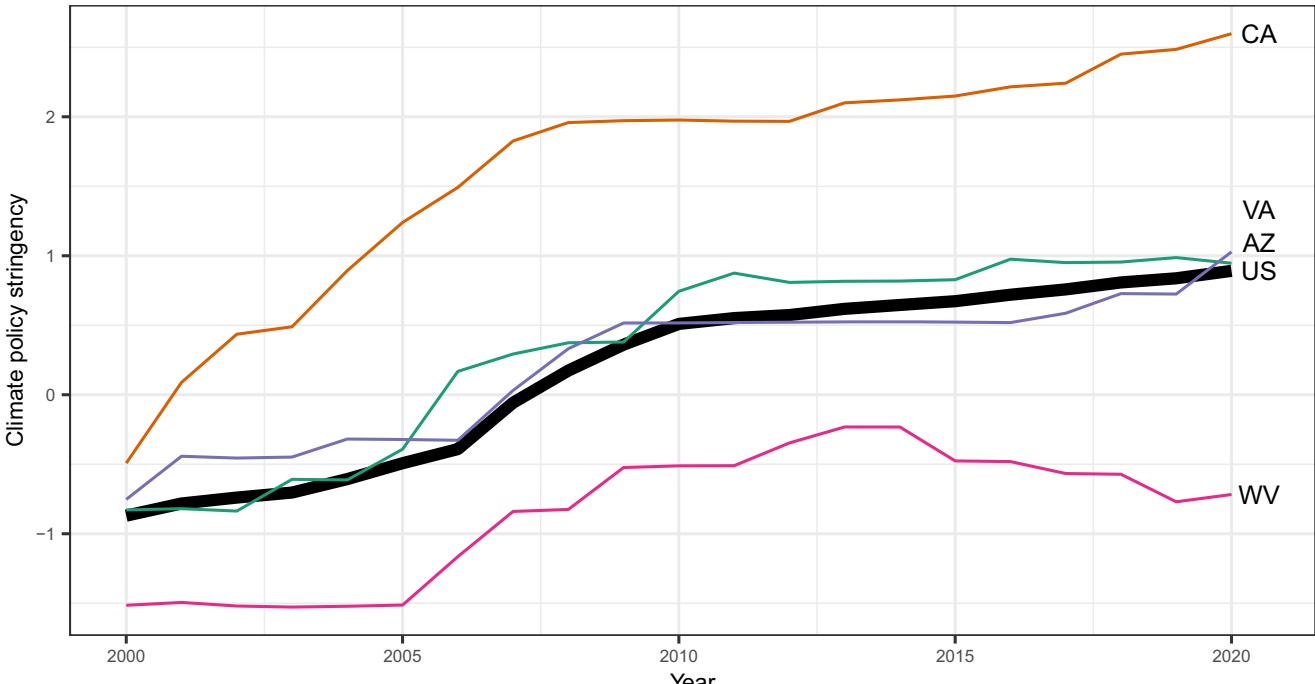

**Fig. 1 | Increasing state climate policy commitments over time.** Each thin line represents the trajectory of an individual case state, and each of these case states is shown in its own color for clarity. The bold line shows the average of our climate policy stringency index across all the states, with each state weighted equally. Supplementary Fig. S2 shows climate policy stringency for all states.

grained distinctions between specific states' policy designs). Of course, our data cannot exhaustively quantify everything that every state is doing to address climate change. For instance, we do not include policies which, for geographic reasons, are only relevant in a handful of states.

We use these policies, coded as ordinal, dichotomous, or continuous, as inputs to a Bayesian factor analysis model[21,22] to estimate the stringency of states' climate policy regimes. Our approach, described in detail in the Methods section, weights the policies according to the information they provide about each state's commitment to a clean-energy transition, and accounts for variation in the stringency of each policy instrument between states. The resulting set of estimates provides a ranking of state climate policy efforts on the dimensions across which states are comparable, from 2000 to 2020 (Fig. 1; See also Supplementary Fig. S2 and Supplementary Table S1, which show the climate policy stringency estimates and relative rankings for each state, respectively.). Our estimation approach reduces the measurement error that stems from limited documentation of policies within many states, failure to incorporate the diversity of policy instruments states are using, coarse comparative coding of policy instruments across states, and a failure to account for differences in distinct policy instruments' contributions to state efforts. We include results from a series of tests of construct validity and convergent validity in the Supplementary Information (Figs. S3, S4, S5).

Our index of state climate policy regimes shows the evolution over time of state-level efforts to mitigate climate change through the energy system. Figure 1 shows climate policy for several illustrative states and the national average, in each year of our time series. The figure shows that all states have taken some action to mitigate climate change, and that states vary widely in the strength and trajectories of their policy commitments. The average state increased its climate policy by 1.76 standard deviations across the full time period included in our data. Cross-sectionally, this is equivalent to the difference between California, at the upper end of the scale in 2020, and Arizona in the same year. It is also equivalent

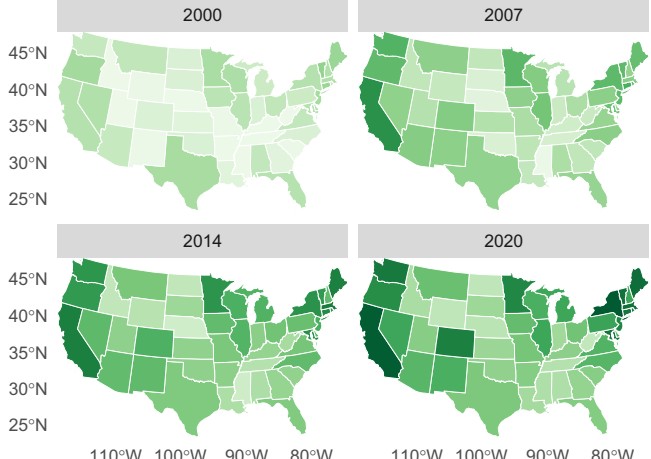

**Fig. 2 | State climate policy, 2000–2020.** States shaded with darker greens have enacted a higher number of stronger climate policies, and states shaded with lighter greens have enacted fewer, weaker climate policies. The maps were developed using the R package Tigris[58] and based on spatial data provided by the US Census Bureau. Supplementary Fig. S1 includes Alaska and Hawaii, and Supplementary Table S1 shows the estimate for each state in 2020.

to the difference, in 2020, between West Virginia, at the lower end of the scale, and Virginia.

Figure 2 delves into this variation, showing which states have led and lagged in enacting policies to promote a clean energy transition. Together, Figs. 1 and 2 show increased initiative from some of the current liberal states, such as New York, California, and Massachusetts. The figures also comport with prior work documenting early leadership followed by a slowdown or retrenchment in states like Texas and Ohio[42]. These patterns illustrate the face validity of our estimates. Additionally, the figures show climate policy adoption in states like

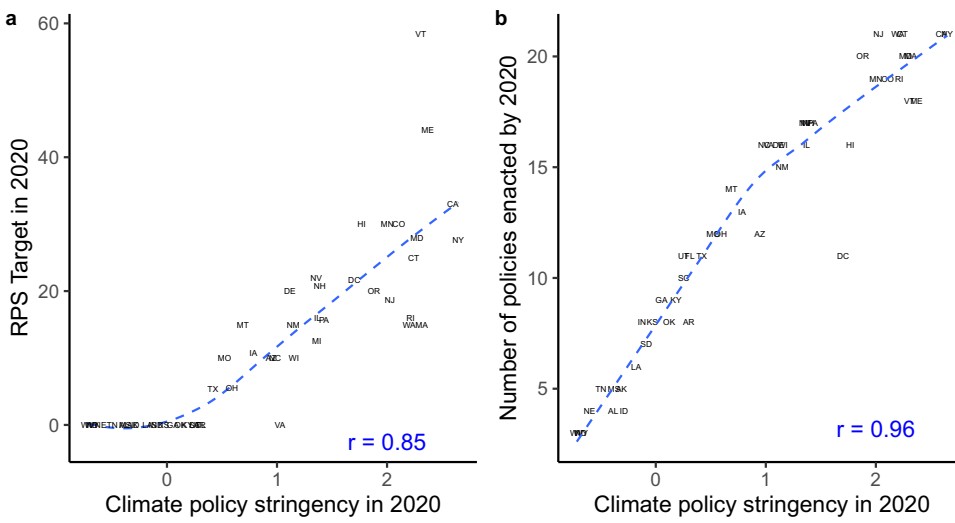

**Fig. 3 | Climate policy index compared with simpler measures of climate policy.** Panel (**a**) shows the relationship between the aggressiveness of states' RPS targets, defined as the percent of utilities' electricity production that must be generated from renewable sources in 2020, and our climate policy index in 2020. Panel (**b**) shows the number of policies that each state had enacted in 2020 and that state's climate policy in 2020. The figure visualizes the variation that our index captures by incorporating numerous policies, estimating discrimination parameters for each policy type, and accounting for variation across states' versions of the same policy instrument. Both panels include the Pearson's R correlation coefficient and locally weighted smoothing (Lowess) line.

Colorado, Minnesota, and Washington, which do not stand out for the liberalism of their broader state policy regimes[43]. These distinctions suggest that climate policy commitments reflect a distinct dimension of policymaking that merits its own measure, apart from broader measures of policy liberalism.

Figure 3 shows how incorporating numerous policies and weighting them according to their relative stringency allows the climate policy index to reflect variation in climate policy regimes that a simple additive index or a focus on any single policy would obscure. Panel (a) shows the relationship between states' RPS targets and our climate policy stringency index in 2020. The left edge of the plot shows that many states have not adopted an RPS; they would all share a value of zero if our measure were RPS adoption or RPS stringency. We detect variation between these states' climate policy regimes because they have adopted various other climate policies. Thus, the plot emphasizes the value of focusing on multiple policies, holistically.

Panel (b) shows the value of accounting for variation in policy design and stringency. It shows the number of policies to address climate change that each state had enacted in 2020, and climate policy stringency in the same year. Note that if our index were a simple count of policies, the points would fall exactly along a linear best-fit curve. Instead, our index captures variation according to which policies have been adopted. For example, Indiana and Arkansas had both adopted eight policies in 2020. Arkansas scores higher on our index because most of the policies it has passed are weighted more strongly in our index, compared with those that Indiana has passed. Our index also captures variation in the stringency of the "same" policy instruments. For example, Alabama and Idaho had both adopted four policies in 2020, including two of the same policies: electric and gas system decoupling, ie disassociating electric and gas utility profits from the sales of electricity and gas. However, Idaho has implemented the most stringent form of decoupling, whereas Alabama's program is not as strong. The ability to distinguish between these states at the lower end of the scale increases analytical leverage for describing and assessing the impact of state climate policy efforts.

## The effects of state climate policy

We next assess whether increases in climate policy reduce emissions of carbon dioxide ($CO_2$), which is one of the greenhouse gases most strongly associated with climate change. It is a primary target of the policies included in our index. In Fig. 4, we explore this link. On the left, panel (a) examines the cross-sectional correlation between $CO_2$ and climate policy in the final year of our data set, and panel (b) on the right shows the correlation between overtime changes in climate policy and $CO_2$ emissions (see Methods for definitions and data sources for $CO_2$ emissions and all other dependent variables used in our analyses). The figure suggests that climate policy is associated with decreases in $CO_2$ emissions, both cross-sectionally and over time.

To formally estimate the effect of climate policy on $CO_2$ emissions, we use time-series cross-sectional OLS regression models. In all models, we include state-fixed effects to control for time-invariant differences across states such as political culture. We also include region-year fixed effects to account for annual shocks that affect all states, such as the COVID-19 pandemic, and over-time trends such as shifting regional economies. In Supplementary Figs. S8 and S9, we present results from regression specifications including additional controls. Since our estimates of climate policy are measured with error, we adjust coefficients to account for this error (See the Methods section for more details about our regression specification and error correction)[25].

Panel (a) of Fig. 5 shows the results from our assessment of the effect of climate policy on $CO_2$ emissions. A standard-deviation increase in climate policy is associated with a 5 percent reduction in annual per-capita $CO_2$ emissions from the electricity sector ($t = -2.3$, error-corrected). We find slightly weaker evidence, in terms of substantive significance, that climate policy helps to reduce emissions more broadly across the economy ($t = -2.06$, error-corrected). Supplementary Table S2 shows coefficient estimates both with and without adjusting for measurement error. The models that do not correct for measurement error have larger coefficients and smaller standard errors, but the general pattern of results is consistent across both sets of models.

We next explore the mechanisms for this effect on $CO_2$ emissions by examining the effect of climate policy on energy production and consumption in each state. Panels (b) and (d) of Fig. 5 show the results from these analyses. The pattern of results suggests that climate policy is associated with approximately 3% reductions in energy and electricity consumption in a state ($t = -2.3$, $-2.0$, $-2.1$, error-corrected), with weaker evidence for reductions in electricity production ($t = -1.875$, error-corrected). We do not detect an effect of climate

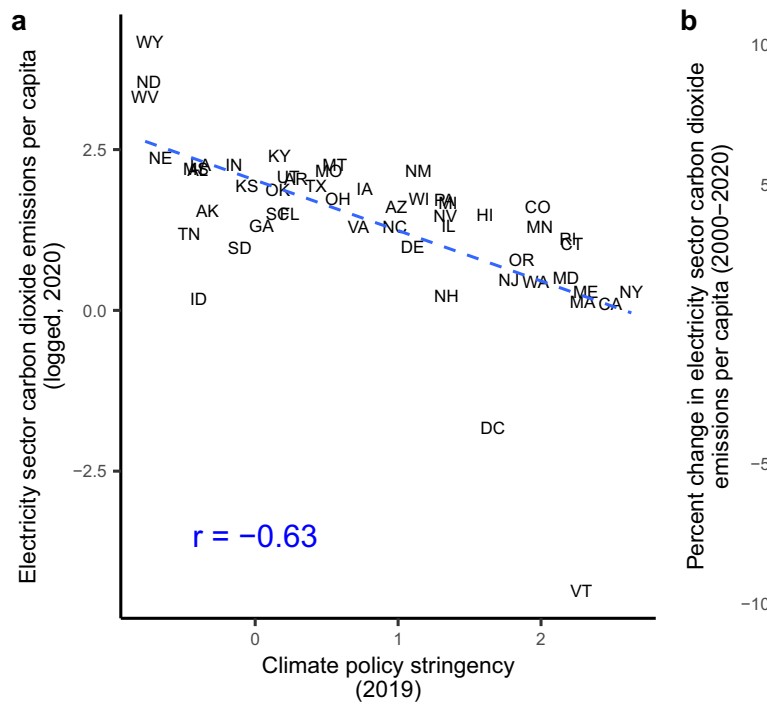

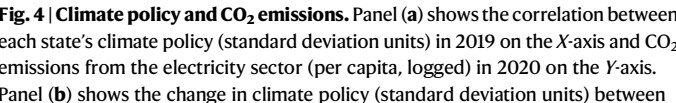

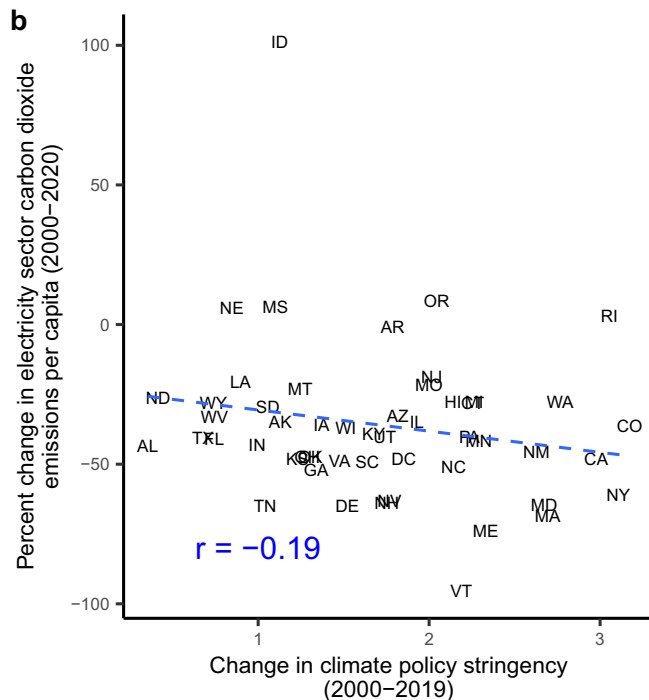

**Fig. 4 | Climate policy and CO₂ emissions.** Panel (**a**) shows the correlation between each state's climate policy (standard deviation units) in 2019 on the *X*-axis and $CO_2$ emissions from the electricity sector (per capita, logged) in 2020 on the *Y*-axis. Panel (**b**) shows the change in climate policy (standard deviation units) between 2000 and 2019 on the *X*- axis and the change in logged per capita $CO_2$ emissions from the electricity sector between 2000 and 2020 on the *Y*-axis. Both panels include the Pearson's *r* correlation coeffient and linear best-fit line. Both panels suggest a strong association between climate policy and $CO_2$ emissions.

policy on some of the other important targeted outcomes, such as renewable energy production (aggregated across sources) or coal-fired electricity production.

We next examine whether climate policy has a detrimental effect on states' economies. Research suggests that communication about economic costs and benefits are crucial determinants of the political viability of clean-energy policies[29,44–46], and political opponents of climate policy often claim that it will raise electricity prices, kill jobs, or stunt economic growth. Extant research has not decisively shown the real economic impacts of climate and energy policies, and the climate policy index opens the opportunity for such an assessment. Panel (c) of Fig. 5 shows the results from models that examine the effect of climate policy on electricity prices, GDP, jobs, and wages in each state. These effects are statistically insignificant at both the $\alpha = 0.05$ and $\alpha = 0.1$ levels ($t = 0.74$, $-1.4$, $-1.5$, $-1.4$). We do not find evidence that climate policy harms the economy.

## Discussion

In this paper, we build on a growing body of research highlighting the importance of comparing climate and energy policies with more nuance than binary indicators or additive scales allow. Most notably, scholars have shown the utility of incorporating design differences into the assessment of RPS policy effectiveness[4,10,13,14]. Here we extend this insight that state policy designs vary in ways that are consequential for assessments of policy impact. We also broaden the scope of analysis to conduct a holistic analysis of states' climate policy regimes, rather than studying policies in isolation. We ask: how have states' climate policy regimes changed over time, and how do changing climate policy regimes affect outcomes that matter for human lives?

Our climate policy index incorporates a wide array of policy instruments that have mostly been studied in isolation. We model the information that each type of policy instrument provides about each state's commitment to a clean-energy transition. Our model also incorporates the variation between states' climate policy instrument

designs. The resulting estimates show how each state's climate policy regime has developed over time, and how states' climate policy regimes compare with each other. We use the index to assess the effects of climate policy on planet-warming pollution, the energy system, and the economy.

We find that a 1 standard-deviation increase in climate policy is associated with a 5 percent decrease in $CO_2$ emissions from the electricity sector and a 2 percent decrease in $CO_2$ emissions across the economy. Thus, state climate policy matters for reducing planet-warming $CO_2$ emissions. However, we do not find that climate policy is associated with an increase in renewable energy production or a reduction in fossil fuel-based energy production in each state. Instead, we find that increasing climate policy is associated with a reduction in electricity consumption overall. These results are consistent with projections that increasingly stringent emissions reduction scenarios would be cost-neutral if climate policy is subnationally heterogeneous, due to energy trading between states[47]. Alternatively, these findings may suggest that climate policy is particularly effective at spurring efficiency improvements within states. Also consistent with the projection of cost-neutrality under a subnationally heterogeneous regime, we do not find evidence that climate policy harms states' economies by killing jobs, depressing wages, stunting GDP growth, or raising electricity prices.

It is certainly good news that climate policy regimes are associated with a measurable reduction in planet-warming $CO_2$ emissions. Nonetheless, the analysis also suggests that current state policies are insufficient to promote the sustained reduction in $CO_2$ emissions required to meet the target set by the Paris Climate Accord: achieving net-zero $CO_2$ emissions and limiting global warming to 1.5 degrees Celsius. Under this agreement, the US needs to reduce emissions by 50 percent by 2050[48]. According to our analysis, the average state increased its climate policy by 1.76 standard deviations across the full time period included in our data. At this rate, our model predicts that the climate policies states have enacted (on average) have reduced state-level per-capita emissions by 9.6 percent over the course of two

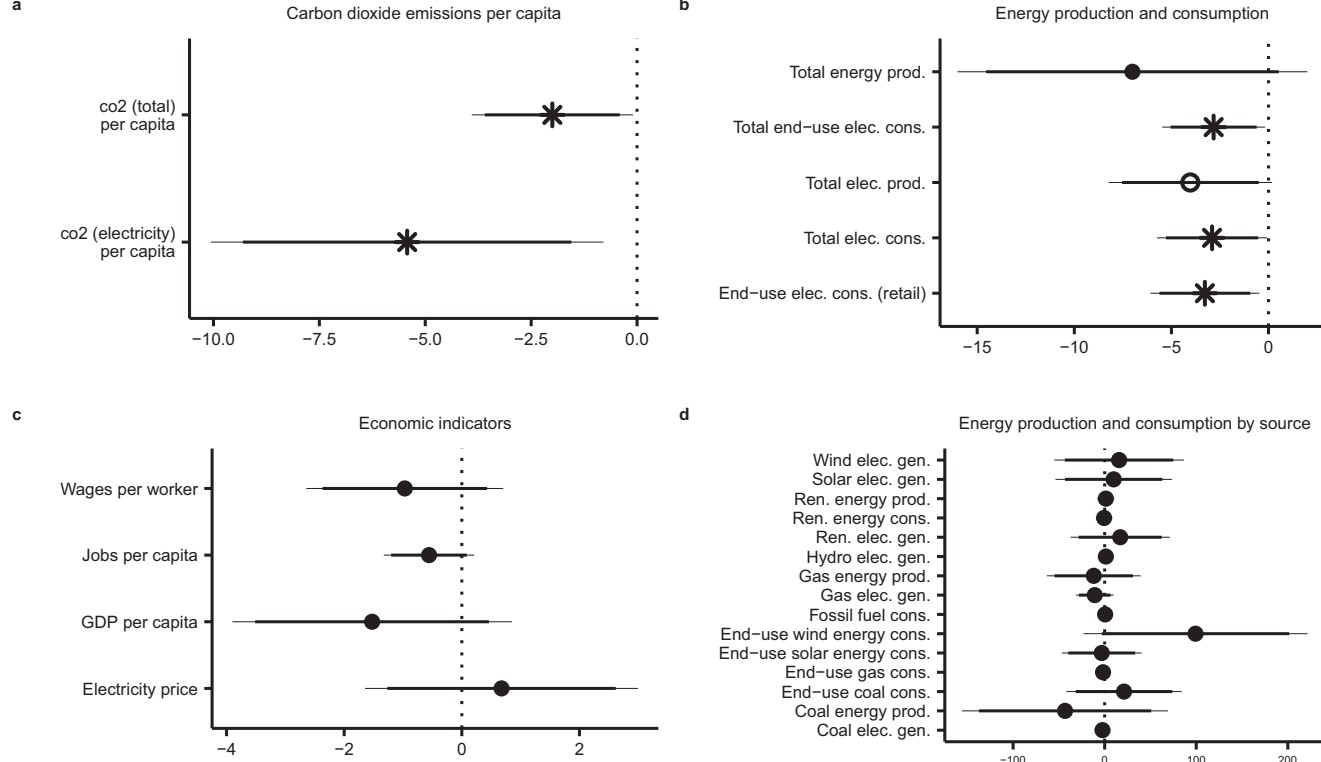

**Fig. 5 | Effects of climate policy on carbon dioxide emissions, energy production and consumption, and the economy.** The figure shows the effect of the climate policy stringency index (on the $X$-axis) on $CO_2$ emissions (panel (**a**)), energy production and consumption (panels (**b**) and (**d**)), and economic indicators (panel (**c**)). We estimate the effects of climate policy on each dependent variable using OLS regression models that include state and region-year fixed effects and standard errors clustered by state and region-year. Each regression is estimated for 1071 observations, including 51 geographic units across 21 years, in 4 regions. Regression coefficients, shown as points in the figure, have been corrected for measurement error in our climate policy index[25]. Thick error bars reflect 90% confidence intervals, and thin error bars reflect 95% confidence intervals. Statistical significance is designated with stars ($\alpha = 0.05$) and hollow points ($\alpha = 0.1$). Climate policy is standardized to have mean of zero and standard deviation of one. All dependent variables are logged, and coefficients have been multiplied by 100 so that the figure approximately shows the percent change in these outcomes associated with a standard-deviation increase in climate policy.

decades. This falls far short of the scale of action needed to meet the goals of the Paris agreement. Moreover, states vary widely in their climate policy regimes. At the low end, West Virginia increased the stringency of its climate policies by 0.8 standard deviations. Conversely, California and New York, at the high end, increased their climate policy efforts by 3 standard deviations.

Our analysis opens a plethora of opportunities for future research. First, the descriptive results presented here raise questions about the drivers of climate policy in the states. For example, Fig. 1 shows a notable uptick in climate policy stringency between 2005 and 2008. In the future, scholars might investigate the state-level and national drivers of this and other broad trends. In particular, scholars might focus on the political drivers of the trends shown in Fig. 1. In this spirit, scholars should also continue to probe deeply the political dynamics that are particular to specific policies contained in our index. Such work provides a crucial complement to our broad approach to policy measurement, which necessarily obscures deep analysis of particular policies including those that are only adopted in a handful of states. Second, the climate policy index might be used to study the impacts of climate policy changes for how the public thinks about energy and climate policy. Third, the climate policy index allows for assessment of the distribution across society of the environmental and economic impacts of climate policy. Fourth, as the climate continues to change and climate policy regimes continue to evolve, the climate policy index will allow for the continued assessment of the environmental and economic impacts of changes in climate policy. We also hope that the climate policy index will spawn a myriad of other creative applications that we have not anticipated.

## Methods
### Data
We compile the climate policy data for our index from advocacy groups, government websites, and academic sources. (See Supplementary Table S3). We began by reviewing published work on this topic and gathering time-series data on every policy that had been included in prior analyses of state climate policies or state policy more broadly. Next, we consulted the websites of several prominent NGOs that aggregate data on state climate policy (e.g., the Database of State Incentives for Renewables & Efficiency[49], the American Council for an Energy Efficient Economy[50], and the National Council of State Legislatures[51]). From these websites we downloaded the policy data or, if the policy data were not available from the NGO, we obtained it from state websites. Next, we supplemented this data with policies that we knew to exist due to our domain area expertise in climate and energy policy. To the extent possible, we code the stringency of each policy relative to versions of the same policy instrument enacted in other states. The coding scheme for each policy is continuous, ordinal, or binary; we use the most granular coding scheme that is feasible based on the nature of the policy, existing cross-state comparisons of the policy, and available data. The Supplementary Information includes further details on the coding, sources, and temporal coverage for each policy variable. For Fig. 3, we use data on RPS targets that were generously shared by Solomon and Zhou[41]. We updated the data to include years not included in Solomon and Zhou's (2021) original analysis.

Our outcome data come from the US Energy Information Administration (EIA) datasets on energy-related emissions of $CO_2$[52]. We use two different dependent variables in our main results. The total

emissions variable reflects emissions from direct fuel use across the residential, commercial, industrial, and transportation sectors, as well as primary fuels used to generate electricity. The electricity-sector emissions variable reflects emissions from the US electric power industry. Emissions are reported in metric tons in the EIA dataset. We use population data from the US Census intercensal population estimates[53] to create per-capita emissions variables. All analyses use the natural log of these variables (with state and region-year fixed effects).

We use data on energy and electricity production, consumption, and price from the EIA. The results shown in Fig. 5 includes measures from several EIA datasets. These include:

- Primary energy production and consumption by source, including the production of fuels used in electricity generation but excluding electricity production (to avoid double-counting)[54]
- Net electricity generation from the electric power industry, by source[52]
- Total energy production and consumption, including all primary energy sources used directly by the residential, commercial, industrial, transportation, and electric power sectors, as well as net interstate flow of electricity and net imports of electricity[54]
- Average annual electricity price (cents per kilowatt-hour)[54].

Production and consumption data are provided in billion British Thermal Units (bBTU). For source-specific variables (eg, solar energy consumption or production), we calculate the proportion of a state's total generation that is attributable to each source. In all regression models, we use the natural log of the energy production and consumption variables.

We use economic data from the Bureau of Economic Analysis[55]. The jobs variable is a count of the total number of full-time and part-time jobs per capita in a given state and year, including wage and salary jobs, sole proprietorships, and individual general partners, but not unpaid family workers nor volunteers. The wages variable reflects the total wages and salaries per worker (in thousands of US dollars) payable by employers to employees during each year in each state. The GDP variable reflects each state's gross domestic product per capita (in million US dollars) in a given year attributable to all private industries and government activity. In our regressions we use the natural log of jobs per capita, GDP per capita, and wages per worker. Union membership data come from Hirsch and Macpherson's database compiled from the Current Population Survey[56]. Supplementary Table S4 provides a description, original units, source, variable names in our analyses, and replication file names for each outcome variable.

## Analysis

The climate policy index is similar in concept to a number of cross-sectional state-level rankings of climate policy effort. The most detailed of these rankings is the scorecards compiled by the American Council for an Energy Efficient Economy (ACEEE). The estimation we undertake is similar in concept to the ACEEE scorecards, but more appropriate for statistical analyses for several reasons. First, we use the same set of policies over time, such that our estimates are suitable for time-series analysis of either a descriptive or a causal nature. Relatedly, we include many policies that the ACEEE scorecards do not include. Second, we let the data determine how to weight each policy, according to the observed relationships between the policies. Third, our index includes only policies, whereas the ACEEE scorecards include some indicators that would more appropriately be considered outcomes in our analysis. This ensures that our index is suitable for use as either a predictor of environmental outcomes or an outcome of the policy-making process. The resulting climate policy stringency index summarizes and ranks state climate policy efforts over time.

To estimate the stringency of states' climate policy regimes, we use a Bayesian modeling approach similar to that used by Caughey and Warshaw (2016)[22]. Our approach builds on Quinn's (2004) Bayesian factor analysis using mixed ordinal and continuous data[21]. Similar to item-response theory models, our approach weights the policies according to the information they provide about each state's commitment to a clean-energy transition. Since many of our inputs are ordinal or continuous, the model also incorporates the variation in stringency of each policy instrument between states. We hold each policy's intercept (difficulty parameter) constant over time, which allows us to compare states' climate policy regimes over time. The policies' discrimination parameters–analogous to slopes reflecting the information each policy contributes to our assessment of states' policy stringency–are also held constant over time[22]. Supplementary Fig. S3 shows the estimated discrimination parameters for each policy in the index. We use diffuse priors and calculate our Bayesian model using the R package `dbmm: dynamic Bayesian measurement models` (https://github.com/devincaughey/dbmm), which uses the Bayesian programming language Stan, as linked to R by the package CmdStanR. We calculate the model by running four chains for 1000 iterations each, including 500 iterations for warm-up and 500 iterations that we save for analysis.

In the Supplementary Information (Supplementary Figs. S4, S5, S6), we assess the convergent validity and construct validity of our estimates. We assess convergent validity by comparing our estimates with the American Council for an Energy-Efficient Economy (ACEEE) state-level energy efficiency scorecards[50]. We find that the climate policy index correlates well with the ACEEE scores, and that the correlation improves over time. To assess construct validity, we compare our estimates with estimates of state policy liberalism[22] and public ideological preferences[57]. We find strong correlations with these measures as well.

We use time-series cross-sectional OLS regression models to estimate the effect of climate policy on emissions, energy production and consumption, and economic indicators. All models regress the natural log of the dependent variable on our scaled (mean=0, std.dev=1) climate policy index. Thus, all coefficient estimates reflect the proportional change in each outcome that is associated with a standard-deviation increase in climate policy. In all models, we include state-fixed effects to control for time-invariant differences across states such as political culture. We also include region-year fixed effects to account for annual shocks that affect all states, such as the COVID-19 pandemic, and over-time trends such as shifting regional economies. In all models we cluster standard errors at the state and region-year level.

We adjust coefficients to account for measurement error in our climate policy index, using an approach similar to bootstrapping called the Method of Composition (MOC)[25,43]. First, it is important to note that since our index is estimated using a Bayesian approach, we estimate a distribution of climate policy for each state-year. We use the median of this distribution in our graphs of climate policy estimates for each state-year and in regressions that are unadjusted for measurement error. To adjust our regression coefficients for measurement error, we draw 100 samples from the distribution of climate policy. For each of these 100 samples from the distribution of climate policy, we estimate our regressions and then take a sample from the distribution of regression coefficients. We use this distribution of regression coefficients to estimate the point estimates and standard errors that we report in the paper. In Supplementary Table S2, we present both unadjusted and measurement-error corrected coefficient estimates.

The Supplementary Information shows several robustness checks. First, we examine the sensitivity of our results to the inclusion or exclusion of any of the policies included in our index. The results are shown in Fig. S6 and show that our results are robust to the specific policies included in the model. The point estimates vary slightly, but

they are almost all statistically significant ($p \leq 0.05$) and similar in magnitude. Second, we examine the robustness of our results to regression specifications including additional controls for lagged economic indicators (Supplementary Figs. S8, S9). The pattern of results is robust to these alternative specifications.

## Reporting summary

Further information on research design is available in the Nature Portfolio Reporting Summary linked to this article.

## Data availability

The data for this study have been deposited in the Harvard Dataverse at https://doi.org/10.7910/DVN/PXWXWI. See Supplementary Tables S3 and S4 for descriptions and sources of our policy data and outcome data. The data reposity includes data from all sources listed in Tables S3 and S4.

## Code availability

The code for this study has been deposited in the Harvard Dataverse at https://doi-org.proxy.library.upenn.edu/10.7910/DVN/PXWXWI.

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

## Acknowledgements
The authors are grateful to Gloria Li and Nathalie Kirsch for data collection assistance; Sanya Carley, Barry Solomon, and Shan Zhou for sharing data on RPS targets and stringency; and Devin Caughey, Jon Ladd, Bill Gormley, Mark Richardson, Josh Basseches, Jack Mewhirter, participants at the 2022 State Politics and Policy conference, participants at the University of Washington's Program on Climate Change Summer Institute, and members of the University of Pennsylvania's political science department for helpful feedback.

## Author contributions
The authors contributed equally to drafting the text of the paper. Both authors gathered data on climate indicators. C.W. had primary responsibility for estimating the models of state climate policy. P.B. had primary responsibility for the descriptive results and regression models in the paper.

## Competing interests
The authors declare no competing interests.
