## [Peer Review File · Nature Communications]

How climate policy commitments influence energy systems and the economy of US StatesREVIEWER COMMENTS

Reviewer #1 (Remarks to the Author):

This article offers a new index of climate policy stringency across US states, and examines the association between this index and climate policy outcomes and economic outcomes. It represents a series of important advances in both our measurement and analysis of climate policy impacts. In general, I am supportive of seeing this piece published in Nature Communication.

There are a few things that are particularly strong in my mind. First, I am glad to see scholars undertake a proper Bayesian analysis of climate policy stringency, and think that the paper's stringency index will become a standard dataset used by many scholars. Second, the descriptive analysis here sets important benchmarks for research on policy efficacy. The finding on the link between stringency and economic growth is also sure to be of interest.

With this said, there are some smaller queries and suggestions I have for the authors to consider.

- I wanted to see a slightly stronger theoretical conceptualization of the latent variable here. While it is, to some degree, policy stringency, it is really more a measure of proactive government support for climate actions? In fact, is this more clearly an index of electricity-system decarbonization stringency?

-Here, the state pre-emption bans stand out as an interesting exception. In fact, this item makes me wonder why the authors have not included many other electricity and climate-related bills that have been passed across states by legislatures working to undermine the energy transition - from setback laws for wind to laws that criminalize protest against pipelines. I do understand that the authors are looking for policies that, in principle, matter for every state. As a result, many policies related to oil and gas extraction / fracking etc. may be difficult to conceptualize. However, this feels like a missing element of the analysis -particularly when the analysis moves beyond electricity sector measurements to consider economy-wide decarbonization trajectories.

- A more minor point - I was confused by the state pre-emption policies were coded as a 1 (when that is the anti-policy position). Relatedly, does Figure 1 include this item? Surely, it should not. I am not entirely sure, when I think about it, whether there is any way to undertake additive analysis of policies with different valences.

- Relatedly, I was less surprised by the difference in results from the electricity sector vs. the economy as a whole, given the specific policies being measured are very electricity-sector in their nature. A lot of state-wide decarbonization is presumably going to be a function of state-specific industrial, transport, and extractive industries.

- This will come out in the Bayesian wash (I think), but it is worth noting the presence of some measurement error in some of the coding here. For instance, my understanding is that some RPS policies also vary in the lists of eligible technologies. If Pennsylvania includes (to my understanding) coal-bed methane in their RPS, does that change how we should think about things?

- Descriptively, it is clear there are sharp increases in policy counts mid-way through the dataset, and clearly visible in Figure 1. It might be interesting to present, in an exploratory fashion, the correlations of these iterative jumps. Do they reflect particular constellations of political factors (e.g. complete control of state policymaking by Democrats?)

Reviewer #2 (Remarks to the Author):

Review of NCC US State Climate Policy Commitments

This manuscript uses coding of various state policy commitments to construct a climate policy stringency index. The face-value checks of the measure and its correspondence with other measures is useful, as is the exploration of how it better captures stringency than other measures used in the literature. Then the manuscript uses the stringency index as a predictor of Co₂, energy production and consumption, and economic indicators to determine whether these policies are associated with reductions in emissions and the mechanism by which that might have happened, as well as whether it came at an economic cost. The exploration of mechanisms is interesting and the discussion nicely draws out the implications and the fact that the states are far from stringent enough to meet climate goals.

I have three sets of concerns that should be addressed.

1) What would make the index more convincing is evidence that the policies included in the dataset are comprehensive. On p. 2, the authors state that they “compile a comprehensive dataset that reflects adoption and design differences across states for each of 21 policies multiple states have used to reduce” GHGs, etc. This sentence could have multiple interpretations: that the authors identified 21 arbitrary policies and compiled comprehensive data about their adoption and designs, that the authors identified all policies that were adopted in more than one state (“multiple” – this interpretation, if correct, is particularly problematic – imagine one state banned GHG emissions outright and did nothing else – it wouldn’t get any credit in this scheme!), or that the authors did indeed compile a comprehensive dataset of ALL policies adopted for these purposes in states. And if this last is what is meant, the authors have an obligation to provide some evidence that it is comprehensive. What is provided in the data section does not give confidence that the policy data are comprehensive, as it simply offers the sources of the data, not the search strategy.

2) The results in Figure 5 come from very minimally specified regressions (e.g., wages as a function of just the stringency index). This would leave them quite subject to omitted variable bias (though the region-year fixed effects help). For example, what if unionization rates are correlated with both wages (+) and stringency (-) -- I’m imagining coal workers unions? That would bias the coefficient on stringency downward. I realize that there is a fairly small n, but regressions with at least the typical controls for a given DV would be more convincing.

3) For this to be a sufficiently large contribution for Nature Climate Change, the index would need to be useful to other researchers. This contribution is an improvement over other measures (but certainly not substantially different from them as Figure 3 shows) and the findings are consistent with other analyses that suggest most policies intended to reduce emissions do so. Is the index useful for other questions? Or does this paper represent its primary usefulness? If it is potentially useful to other researchers, the discussion section should articulate some of the unanswered (or poorly answered) questions that the index could be used to answer. Such a discussion would help the manuscript to reach a broad audience that can use the new index.

Smaller issues:

What Figure 5 represented (the stringency index as a predictor of Co₂, energy production and consumption, and economic indicators to determine whether these policies are associated with reductions in emissions and the mechanism by which that might have happened, as well as whether it came at an economic cost) was very hard to figure out. Because readers are used to seeing independent variables in this kind of figure, the authors should more clearly state the specifications they’re using (something like the parenthetical in the prior sentence would have helped).

Although the authors stated that they used the MOC estimates combined with the point estimates in

their interpretation of the results, I was unable to identify that language in the manuscript. Footnote 3 indicates that they will use the words strongest and suggestive in particular instances, but the text does not seem to indicate when the two estimates both had $\alpha < 0.05$, etc.

The authors should name the index and remove all references to "our measure", instead referring to the name of the index. Perhaps something like Policy Stringency index, as that is occasionally used in the manuscript.

Unclear antecedent on p. 4: "We use Bayesian factor analysis with these...inputs" To what is these referring? The vague word inputs also doesn't help here. Policies? Policy stringency codings?

p. 5: Conform with should be confirm or comport with.

Carbon dioxide should not be capitalized.

Reviewer #3 (Remarks to the Author):

What the authors are attempting to do in this manuscript is very meritorious. They are absolutely correct that having a comprehensive measure of state-level climate policy stringency that considers policy design complexities and is not merely an additive index would be enormously valuable for future research and I applaud their ambitious undertaking to create one. They have clearly put a lot of thought and work into developing this measure, and I applaud them for that, too. Furthermore, all of the analyses performed using OLS regression models are sound, and they reach some interesting and important conclusions about the causal relationships they explore in these analyses.

However, I am afraid that, despite their best efforts, there are some problems with their comprehensive policy stringency measure, the components of which are explained in Table 1.

Firstly, and before I get into some of the issues with their operationalizations, we need to know the data source for each and every policy measure presented in Table 1. Perhaps I missed it, but all I could find was a single sentence in Section 3.1 (p. 13) that says the data are from "advocacy groups, government websites, and academic sources." We need to know what each of these sources are.

Turning now to the component policies and issues with their operationalization and the authors' conceptualization of stringency:

Environmental Protection Act:

What does "local/private" mean?

Utility deregulation:

This should not be a dichotomous variable, and it probably shouldn't even be ordinal (though ordinal would be preferable to dichotomous). It's going to be very difficult to come up with a quantitative measure for this. Qualitative research is likely to be needed to tease out the effects of particular deregulation designs on particular climate and renewable energy policy outcomes. Utility deregulation policies are enormously complex and qualitatively heterogenous. For instance, Texas (which deregulated in 1999 – and is possibly the single most fully deregulated state in the country – but is not listed as a "state adopting [deregulation] by year 1" in Table 1) has deregulated in a qualitatively different manner from (and to a different degree than) Pennsylvania, which did so in a different way from (and to a different degree than) Massachusetts, etc. Other states, like California, which deregulated in 1996 but then "re-regulated" (only partially) in response to the 2001 electricity crisis, are missing from Table 1 altogether. There are distinctions *with* a difference having to do with which

aspects of electricity provision are deregulated—wholesale versus retail, generation versus transmission versus distribution, etc. In fact, “deregulation” is somewhat of a misnomer since public utility commissions continue to have a regulatory role even in fully “deregulated” states, it’s just the nature of that role that changes. “Restructuring” is a preferable term for describing these types of policy changes.

In any event, it was not clear which of the dichotomous values the authors use in their current measure is considered preferable or “more stringent” in terms of their overall DV of climate policy stringency?

One possibility the authors might consider here is just excluding this particular policy from the overall policy stringency measure, because there isn’t a lot of convincing evidence that deregulation helps or hinders climate policy *overall.* The effects can be mixed and highly contingent. So I think a comprehensive stringency measure that simply excluded deregulation altogether, while incomplete, would be fully acceptable and would still offer a great improvement and contribution to the literature over current measures that focus on just one or two policies or that are “coarsely additive,” as the authors correctly argue.

Renewable Portfolio Standard:

Though we know a lot more in the literature about these policies than we do about deregulation/restructuring, once again, the authors’ current conceptualization of RPS policy design/stringency is overly simplistic. Missing are key design elements such as resource eligibility (see Glenna and Thomas 2010), geographic restrictions and accounting rules (see Rountree 2019) and unbundled/tradable versus bundled RECs (see Yin and Powers 2010, which the authors, to their credit, do cite).

In general, I recommend Fischlein and Smith (2013) for a more complete treatment of RPS policy design elements and their effects.

I have some limited concerns about some of the other policies included in Table 1, too, but these may be alleviated if the authors were to provide source material for all of this information.

A few other points:

It would be good for the authors to explain why they chose the time period they did (2000-2020).

On p. 2, the authors discuss the “speed and extent of the technological transformation,” but there are also economic transformations that are relevant, too, and a lot also depends on regulatory discretion. The political economy of each state is critical.

On p. 4, the authors mention “electricity market restructuring (ie, decoupling)” but these are NOT the same thing. Decoupling is an energy efficiency policy that divorces electricity rates from volume of sales whereas restructuring is what the authors seem to refer to in this manuscript as “deregulation.”

References Mentioned Above:

Fischlein and Smith (2013). “Revisiting Renewable Portfolio Standard Effectiveness: Policy Design and Outcome Specification Matter.” *Policy Sciences* 46(3):277-310.

Glenna and Thomas (2010). “From Renewable to Alternative: Waste Coal, the Pennsylvania Alternative Portfolio Standard, and Public Legitimacy.” *Society and Natural Resources* 23(9):856-871.

Rountree (2019). “Nevada’s Experience with the Renewable Portfolio Standard.” *Energy Policy*

129:279-291.

Yin and Powers (2010). "Do State Renewable Portfolio Standards Promote In-State Renewable Generation?" *Energy Policy* 38(2):1140-1149.

Memo for the Reviewers of “US State Climate Policy Commitments, the Energy System, and the Economy”

We appreciate the opportunity to revise and resubmit our manuscript “US State Climate Policy Commitments, the Energy System, and the Economy.” Overall, the reviewers expressed enthusiasm for the approach we take in the paper and its contribution to the literature about climate policy impacts. They also raised a number of questions about the conceptual validity and empirical robustness of our data collection and analytical methods. In response to the reviewers’ helpful comments, we have made the following changes, which we believe greatly improve the manuscript:

- We have bolstered and clarified the conceptual underpinnings of the climate policy index. We do this through an additional figure in the Supplemental Information, a more thorough explanation of our search for policy data, a systematic review of previously published analyses that incorporated multiple climate and energy policies, the addition of several additional policies to the index, and several footnotes that explain the composition of the index and its relationship to state climate policy efforts.
- We have conducted a number of checks to scrutinize the robustness of our measurement approach and our regression specifications. We estimate a series of climate policy indices, each of which excludes one of the policies included in the index, and show that our regression results are robust to the policy components of the index. We also show that our regression results are robust to the inclusion of additional control variables.
- We have clarified the contribution of our paper by highlighting specific ways in which the index could be used by other scholars. In this discussion, we have focused attention on the descriptive findings presented in our paper since these descriptive findings raise important questions that merit future research.
- We have clarified the communication of our findings, both in our figures and in the text.
- We have streamlined the introduction, to highlight our contribution more clearly at the beginning of the paper.

We respond to the reviewers’ questions and suggestions in detail below.

1 Reviewer #1

Reviewer #1 had a positive view of the paper and its contribution, highlighting in particular the value of our Bayesian modeling approach and the substantive importance of the findings regarding policy effects. The comments from Reviewer 1 highlighted the need for a more detailed conceptual explanation and justification of our modeling approach. We have addressed these comments in the Methods section, through several footnotes in the text, and with an additional figure in the Supplemental Information. We provide detailed responses to the reviewer’s suggestions below.

1.1 Conceptualization of the latent variable

I wanted to see a slightly stronger theoretical conceptualization of the latent variable here. While it is, to some degree, policy stringency, it is really more a measure of proactive government support for climate actions? In fact, is this more clearly an index of electricity-system decarbonization stringency?

We thank the reviewer for this question and we have streamlined the Introduction to more clearly conceptualize the index for readers. As we describe in the Introduction, the latent variable we estimate represents a comparative ranking of states' climate policy regimes, conceptualized as the holistic bundle of state climate policy actions. As such, the index is indeed a measure of proactive government support for climate action, as embodied in the policies that each state has enacted. We have tried to include all policies that could affect CO₂ emissions, that are eligible for adoption in all states, and that can be systematically coded for all states. We further elaborate on this conceptualization in the Methods section (page 18).

The reviewer aptly notes that the index focuses more heavily on electricity-system policies, compared with policies that target other sectors. This focus reflects the reality that the electricity sector has received the bulk of states' climate policy effort, relative to other sectors. We elaborate on our evaluation of the comprehensiveness of the index in Sections 1.4 and 2.1 of this memo.

1.2 Universe of policies in the index

In fact, this item makes me wonder why the authors have not included many other electricity and climate-related bills that have been passed across states by legislatures working to undermine the energy transition - from setback laws for wind to laws that criminalize protest against pipelines. I do understand that the authors are looking for policies that, in principle, matter for every state. As a result, many policies related to oil and gas extraction / fracking [sic] etc. may be difficult to conceptualize. However, this feels like a missing element of the analysis -particularly when the analysis moves beyond electricity sector measurements to consider economy-wide decarbonization trajectories.

As the reviewer mentions, the conceptual validity of our index rests on its reflecting policies that are applicable in all or most states. Simultaneously, we agree with the reviewer that it would be useful to include policies that are “reverse coded,” ie that reflect efforts to undermine rather than to advance the clean energy transition. To identify more of these policies, we dove back into the prior literature and the NGO databases from which we gathered our original data. (Table 1 in Section 2.1 of this memo shows the policies included in our index and those included in prior published studies). While this search did spur us to expand the dataset of policies, unfortunately we were not able to identify other broadly applicable and well-documented policies that reflect efforts to undermine the energy transition.

1.3 Advantages of Bayesian model over an additive model

A more minor point - I was confused by the state preemption policies were coded as a 1 (when that is the anti-policy position). Relatedly, does Figure 1 include this item? Surely, it should not. I am not entirely sure, when I think about it, whether there is any way to undertake additive analysis of policies with different valences.

The reviewer aptly notes here that we could not include policies intended to undermine—rather than advance—a clean energy transition, if we were estimating a simple additive index. But the Bayesian measurement model for our climate policy index can include policies with the opposite (ie, anti-climate) valence. The model simply assigns these types of policies a negative discrimination parameter (as shown in Figure 1, in this memo). We view the ability to include reverse-coded policies, such as a preemption against local governments’ ability to ban gas hookups, as a strength of our approach vis a vis a simpler additive model.

Our estimates of latent climate policy stringency can include policies that both support and undermine a clean-energy transition. As such, Figure 1 in the manuscript does include state preemption against local gas bans, an anti-climate policy.

A major distinction between our model and an additive index is that our model estimates a discrimination parameter for each policy. This parameter is analogous to a slope, indicating the degree and direction of change in the climate policy index that is associated with the adoption of each policy input. For policies that are negatively associated with the other policies in the model—as is state preemption of local gas bans—the model estimates a negative discrimination parameter. To clarify this for readers, we have added Figure 1 (in this memo) to the Supplementary Information as Figure S1. The figure includes a caption which explains the meaning of the discrimination parameter.

Figure 1: **Discrimination parameters for the policies included in the climate policy index:** Discrimination parameters of higher magnitude indicate that the policy is strongly related to our latent climate policy index, whereas discrimination parameters with values close to zero are weakly related to the latent climate policy index. Positive discrimination parameters indicate that the adoption of the policy is associated with an increase in the latent climate policy index, whereas negative discrimination parameters indicate that the adoption of the policy is associated with a decrease in the latent climate policy index.

1.4 Focus on electricity sector in the index

Relatedly, I was less surprised by the difference in results from the electricity sector vs. the economy as a whole, given the specific policies being measured are very electricity-sector in their nature. A lot of state-wide decarbonization is presumably going to be a function of state-specific industrial, transport, and extractive industries.

We agree with the reviewer that the stronger result on electricity-sector emissions is unsurprising, since many of the policies in our climate policy index focus on the electricity sector. We worked to make the index as holistic as possible, in the ways that we describe thoroughly in Section 2.1. However, the reality is that the electricity sector has received the bulk of states' climate policy effort, relative to other sectors. This is largely because the (relatively) centralized electricity sector is (relatively) easy to decarbonize and because of the large number of solutions available (as reflected in our index). Efforts to decarbonize other sectors are comparatively new, diffused (and thus difficult to measure), and less comprehensive. We have noted this in the text (footnote 3 on page 3), when we introduce Table 1. It should also be noted that to ensure that our estimates would produce a valid comparison between states, we did not include policies that are only applicable in a small number of states.

1.5 Measurement error in policy coding

This will come out in the Bayesian wash (I think), but it is worth noting the presence of some measurement error in some of the coding here. For instance, my understanding is that some RPS policies also vary in the lists of eligible technologies. If Pennsylvania includes (to my understanding) coal-bed methane in their RPS, does that change how we should think about things?

We have added footnote 4 on page 6 to highlight for readers that even as our coding reflects the most granular possible distinctions between states and over time, it is not feasible to perfectly capture all of the distinctions due to data limitations. The benefit of the index is that, by pooling information across many measures, we produce a set of estimates that is more comprehensive than any single-policy measure would be.

1.6 Descriptive inference

Descriptively, it is clear there are sharp increases in policy counts mid-way through the dataset, and clearly visible in Figure 1. It might be interesting to present, in an exploratory fashion, the correlations of these iterative jumps. Do they reflect particular constellations of political factors (e.g. complete control of state policymaking by Democrats?)

We agree with the reviewer that this and many other aspects of the descriptive findings in Figure 1 merit exploration. Indeed, the comment dovetails with the suggestion from Reviewer 2 that we highlight the ways in which our index could be used by other researchers. Examining the drivers of this uptick is beyond the scope of this paper, but we have noted it as an example of an opportunity for future research, in the Discussion section.

2 Reviewer #2

Reviewer 2 lauded the index as an improvement over prior measures of climate policy, echoed the first reviewer’s concerns about our claims that the index is comprehensive, and pushed us to specify how the index could be used by other scholars. The reviewer also raised concerns about the specification and communication of our regression results and made several stylistic suggestions. We address these comments in detail below.

2.1 Comprehensiveness of the index

What would make the index more convincing is evidence that the policies included in the dataset are comprehensive. On p. 2, the authors state that they “compile a comprehensive dataset that reflects adoption and design differences across states for each of 21 policies multiple states have used to reduce” GHGs, etc. This sentence could have multiple interpretations: that the authors identified 21 arbitrary policies and compiled comprehensive data about their adoption and designs, that the authors identified all policies that were adopted in more than one state (“multiple” – this interpretation, if correct, is particularly problematic – imagine one state banned GHG emissions outright and did nothing else – it wouldn’t get any credit in this scheme!), or that the authors did indeed compile a comprehensive dataset of ALL policies adopted for these purposes in states. And if this last is what is meant, the authors have an obligation to provide some evidence that it is comprehensive. What is provided in the data section does not give confidence that the policy data are comprehensive, as it simply offers the sources of the data, not the search strategy.

Thanks to Reviewer 2 for the suggestion that we clarify our search strategy and defend the claim that our policy dataset is comprehensive. We address this suggestion in two ways. First, in the paper, we have clarified that our index does not exhaustively quantify everything that every state is doing to address climate change. Instead it provides a holistic ranking of state climate policy efforts on the dimensions across which states are comparable. The purpose of the index is to reduce the measurement error that stems from poor documentation of policies within many states, failure to incorporate the diversity of policy instruments states are using, coarse comparative coding of policy instruments across states, and a failure to account for differences in distinct policy instruments’ contributions to state efforts. We have gathered as much policy data as possible, while walking back the claim that our measure is exhaustively comprehensive.

We have made this clarification, in two places. First, when we introduce our descriptive results on page 6, we have clarified that our estimates provide a ranking of state climate policy efforts on the dimensions across which states are comparable and highlight the ways in which our coding and measurement model provide an advance over simpler measures. Second, in the Conclusion, we have added footnote 11 on page 14, which again clarifies that the index does not include every policy that every state has enacted but instead provides a ranking of states on the policies across which they are comparable.

Additionally, so that readers can assess the comprehensiveness of our index for themselves, we have added a more detailed explanation of our search strategy to the Methods

section (page 16). We gathered data from prior published studies, NGO websites that aggregate data on state climate policy, and state websites. We supplemented data from aggregated sources with policies that we knew to exist due to our domain area expertise in climate and energy policy.

Table 1 below shows the policies included in prior work and in our index. Our index includes the majority of policies that have been included in prior work, along with many additional policies. Specifically, as the table shows:

- In total, our index incorporates information about 26 distinct climate and energy policies. The most comprehensive prior study on this topic included 11 climate and energy policies (Martin and Saikawa 2017).
- We have included six policies that no prior aggregated study of climate and energy policy has included.
- We have added six policies (shown in red in Table 1) that were not included in our original index and for which we were able to obtain complete time-series data for all or most states.
- We have included 20 of the 24 policies that prior studies have included. We made an effort to recover the data used in each prior study of energy and climate policy, but we were unable to locate data for a few policies. Specifically:
 - Energy efficiency tax credits and corporate renewable energy tax credits: We could not replicate the data obtained by Matisoff and Edwards (2014) from the Database of State Incentives for Renewables & Efficiency (DSIRE) (NC Clean Energy Technology Center 2022). The Matisoff and Edwards (2014) paper used the date of enactment for each policy, whereas our index requires a full history of each policy. We were not able to recover this history for these policies.
 - Personal renewable energy tax credit: We have included an indicator for solar tax credits, which is, in practice, nearly the same as the renewable energy tax credit.
 - Green power purchasing: We could not replicate the data obtained by Shrimali, Lynes, and Indvik (2015) and found the data in DSIRE that is pertinent to these programs to be unreliable. For instance, some programs labeled as “green power purchasing programs” were actually clean energy standards, without evidence of a green power purchasing program contained therein.

Policy in our index	Studies of multiple energy and climate policies							
	Current study	Martin and Saikawa (2017)	Shrimali, Lynes, and Indvik (2015)	Caughey and Warshaw (2022)	Yi (2015)	Prasad and Munch (2012)	Menz and Vachon (2006)	Matisoff and Edwards (2014)
CA Car Emissions Standard	✓			✓				
Climate action plan	✓	✓						
Community solar program	✓							
Complete streets policy	✓							
Electric decoupling	✓	✓						
Emissions performance standards	✓	✓						
Energy efficiency tax credits (corporate)								✓
Energy efficiency tax credits (personal)								✓
Energy efficiency resource standard	✓				✓			
Energy efficiency target	✓	✓						
Environmental Protection Act	✓			✓				
Fuel generation disclosure	✓					✓	✓	
Gas tax	✓			✓				
Gas decoupling	✓							
Greenhouse gas registry/reporting	✓	✓						
GHG target	✓	✓						
Greenhouse Gas Cap	✓	✓		✓				
Green power purchasing			✓					
Low-income energy efficiency programs	✓							
Mandatory green power option	✓	✓	✓			✓	✓	
On-site renewable generation	✓					✓		✓
Preemption of local gas bans	✓							
Property Assessed Clean Energy authorization	✓							
Public Benefit Fund	✓	✓	✓	✓	✓	✓	✓	✓
Public building energy standards	✓							✓
Renewable energy tax credits (corporate)								✓
Renewable energy tax credits (personal)								✓
Renewable portfolio standard enactment	✓	✓	✓	✓	✓	✓	✓	✓
Renewable portfolio standard target	✓		✓					
Solar Tax Credit	✓			✓				
Utility deregulation	✓	✓		✓		✓	✓	

Table 1: **Comparison of our index with previously published energy and climate policy aggregations:** The table shows each policy included in our index and shows a check mark to indicate the inclusion of each policy in each study (indicated as columns), including the present study. Policies shown in red have been added since the original submission.

2.2 Regression specification

The results in Figure 5 come from very minimally specified regressions (e.g., wages as a function of just the stringency index). This would leave them quite subject to omitted variable bias (though the region-year fixed effects help). For example, what if unionization rates are correlated with both wages (+) and stringency (-) – I’m imagining coal workers unions? That would bias the coefficient on stringency downward. I realize that there is a fairly small n , but regressions with at least the typical controls for a given DV would be more convincing.

The reviewer raises a good question about the robustness of our results to model specification decisions. We use a sparse time-series cross-sectional regression specification because it is standard practice in time-series econometric models to use fixed effects to control for time-varying and time-invariant unobservable confounders rather than including control variables that could be post-treatment. Nonetheless, we appreciate the reviewer’s question and respond to it with two robustness checks. First, we add lagged economic indicators to our main specification, with the CO₂ emissions dependent variables. The results are shown in Figure 2 here, and in Figure S6 in the paper. The direction, magnitude, and precision of the results are consistent with the main results presented in the paper.

Figure 2: **Results with controls for lagged economic indicators:** The figure shows the results from our main regression specification including state and region-year fixed effects (left panel), and results from regression specifications including state and region-year fixed effects along with one-year lags for GDP per capita, wages per worker, and jobs per capita.

Second, we run a robustness check in which we add lagged unionization rate to our models assessing the effect of climate policy on economic indicators. These results are shown in Figure 3 here and in Figure S7 in the paper. In this case, some of the coefficients are different in sign, but they remain statistically insignificant.

Figure 3: **Results with controls for lagged unionization rate:** The figure shows the results from our main regression specification including state and region-year fixed effects (left panel), and results from regression specifications including state and region-year fixed effects along with one-year lags for unionization rates in each state.

We have added both of these robustness checks to the Supplementary Information and flagged them for readers in two places: Footnote 8 on page 11, and page 19 in the Methods section.

2.3 Future applications of the climate policy index

For this to be a sufficiently large contribution for Nature Climate Change[sic], the index would need to be useful to other researchers. This contribution is an improvement over other measures (but certainly not substantially different from them as Figure 3 shows) and the findings are consistent with other analyses that suggest most policies intended to reduce emissions do so. Is the index useful for other questions? Or does this paper represent its primary usefulness? If it is potentially useful to other researchers, the discussion section should articulate some of the unanswered (or poorly answered) questions that the index could be used to answer. Such a discussion would help the manuscript to reach a broad audience that can use the new index.

We thank Reviewer 2 for the suggestion that we clarify the novelty of our index and the possibilities it opens for future research. We have added a more detailed link between our index and several possible future research trajectories, in the Discussion section (page 15). This discussion now highlights how the descriptive patterns presented in the paper invite future inquiry into the drivers of climate policy.

We respectfully disagree with the Reviewer’s interpretation of Figure 3 as showing that our index is not substantially different from prior measures of climate policy. Instead, Figure 3 shows our measure to be distinct from prior measures in important ways. First, the left-hand panel of Figure 3 shows that we are able to detect substantial variation between the large number of states that do not currently have a renewable portfolio standard. Were we to focus only on RPS policies, we simply would not be able to say anything about these states, whereas our measure provides a descriptive sense of the differences between them. Measuring the variation between these states also increases the power of any analysis that we run with our measure. Thus, the index increases analytical leverage in both a descriptive and a causal sense.

The right-hand panel of Figure 3 shows that weighting different policies, and distinguishing between states’ versions of the same policies, similarly improves descriptive nuance and analytical power, compared with a simple additive index. For example, if we were using a simple additive index, North Carolina and Hawaii would both have a value of 16 in 2020, but our index distinguishes between the types of policies these two states have adopted and thus reveals that Hawaii’s climate policy regime is more stringent than North Carolina’s.

Finally, we respectfully disagree with Reviewer 2 that the bulk of prior work shows that most climate policies are effective. Table 2 shows recent studies that have analyzed the effects of state climate and energy policies on CO₂ emissions or renewable energy production. The top row of the table shows that, on the whole, studies of RPS have indeed shown that RPS is effective. However, the bottom row of the table shows that results are more mixed from studies that have included multiple policies or any policy other than RPS. This includes two studies that incorporate multiple policies and find that some policies are effective whereas others are not. In these studies, disentangling independent effects is difficult due to likely multicollinearity between policies which are often adopted in combination.

Independent Variable	Significant effects	Null or wrong direction
RPS	Anguelov and Dooley (2019), Barbose et al. (2016), Carley et al. (2018), Menz and Vachon (2006), Pastor (2020), Yi (2015), Yin and Powers (2010)	Bowen, Park, and Elvery (2013), Carley (2009), Zhou and Solomon (2020)
Other policies	Martin and Saikawa (2017), Shrimali, Lynes, and Indvik (2015), Yi and Feiock (2014)	Martin and Saikawa (2017), Menz and Vachon (2006), Prasad and Munch (2012), Shrimali, Lynes, and Indvik (2015), Yi (2015)

Table 2: Prior research assessing the effect of climate and energy policies on CO₂ emissions or renewable energy production

2.4 Clarity of visualization

What Figure 5 represented (the stringency index as a predictor of Co2, energy production and consumption, and economic indicators to determine whether these policies are associated with reductions in emissions and the mechanism by which that might have happened, as well as whether it came at an economic cost) was very hard to figure out. Because readers are used to seeing independent variables in this kind of figure, the authors should more clearly state the specifications they're using (something like the parenthetical in the prior sentence would have helped).

We thank the reviewer for pointing out that the caption below this figure was unclear. We have modified the caption to clarify that the X axis corresponds to point estimates and confidence intervals for the effect of climate policy stringency, and to indicate the dependent variables contained in each panel.

2.5 Statistical significance

Although the authors stated that they used the MOC estimates combined with the point estimates in their interpretation of the results, I was unable to identify that language in the manuscript. Footnote 3 indicates that they will use the words strongest and suggestive in particular instances, but the text does not seem to indicate when the two estimates both had $\alpha < 0.05$, etc.

We have included t statistics for the error-corrected and, in only one case, unadjusted results to clarify this point. As we write in Footnote 9 (previously Footnote 3), the unadjusted results are more precise than the error-corrected results. Thus, readers can assume that in cases where the error-corrected result is statistically significant, the unadjusted results are also statistically significant. Moreover, in the Supplementary Information we have included a full table of error-corrected and unadjusted regression results for readers to inspect themselves.

2.6 Stylistic and grammatical suggestions

- *The authors should name the index and remove all references to “our measure”, instead referring to the name of the index. Perhaps something like Policy Stringency index, as that is occasionally used in the manuscript.*

We thank the reviewer for pointing out the inconsistency in the ways we refer to the climate policy index. We have changed the manuscript so that we call it the “climate policy stringency index” or “climate policy index” throughout.

- *Unclear antecedent on p. 4: “We use Bayesian factor analysis with these... inputs” To what is these referring? The vague word inputs also doesn't help here. Policies? Policy stringency codings?*

We thank the reviewer for noting that our language was unclear here. We have adjusted the text accordingly, to read: “We use these policies, coded as ordinal, dichotomous,

or continuous, as inputs to a Bayesian factor analysis model to estimate the stringency of states' climate policy regimes.”

- *p. 5: Conform with should be confirm or comport with.*

We have changed this sentence to read “comport with.”

- *Carbon dioxide should not be capitalized.*

We have made this correction. Thanks to the reviewer for noticing this mistake.

3 Reviewer #3

Reviewer 3 is enthusiastic about our endeavor to measure climate policy stringency with a Bayesian index, the analyses we use to estimate the effects of climate policy on emissions outcomes, and the overall contribution of the paper. The reviewer raises concerns about the depth with which we have treated each of the indicators in our climate policy index. We thank the reviewer for a positive response to the paper’s goals and to our empirical examination of the effects of climate policy. We respond to the Reviewer’s comments in detail below.

3.1 Overall approach

What the authors are attempting to do in this manuscript is very meritorious. They are absolutely correct that having a comprehensive measure of state-level climate policy stringency that considers policy design complexities and is not merely an additive index would be enormously valuable for future research and I applaud their ambitious undertaking to create one. They have clearly put a lot of thought and work into developing this measure, and I applaud them for that, too. Furthermore, all of the analyses performed using OLS regression models are sound, and they reach some interesting and important conclusions about the causal relationships they explore in these analyses. However, I am afraid that, despite their best efforts, there are some problems with their comprehensive policy stringency measure, the components of which are explained in Table 1.

We acknowledge that our index sacrifices some depth in its treatment of each policy, as a necessary tradeoff against the breadth of our index, in covering many policies over time and across all the states and the District of Columbia. As described in our response to Reviewer 1, we now acknowledge this tradeoff in footnote 4 on page 6, when we describe our data collection, coding, and analytical approach.

3.2 Data sources

Firstly, and before I get into some of the issues with their operationalizations, we need to know the data source for each and every policy measure presented in Table 1. Perhaps I missed it, but all I could find was a single sentence in Section 3.1 (p. 13) that says the data

are from “advocacy groups, government websites, and academic sources.” We need to know what each of these sources are.

The reviewer makes an entirely reasonable request for full source material about each policy included in our dataset. This information is included in a supplementary appendix which we thought we had included in the original submission. We regret if we failed to upload it with the submission. We have incorporated this appendix into the Supplementary Information (Table S2).

3.3 Operationalization of policies

Turning now to the component policies and issues with their operationalization and the authors’ conceptualization of stringency:

3.3.1 Environmental Protection Act

What does “local/private” mean?

Thanks to the reviewer for raising this question. This refers to requirements that local governments or private entities, in addition to state government agencies, submit environmental impact reviews. We have clarified this in Table 1 in the manuscript.

3.3.2 Utility deregulation

*This should not be a dichotomous variable, and it probably shouldn’t even be ordinal (though ordinal would be preferable to dichotomous). It’s going to be very difficult to come up with a quantitative measure for this. Qualitative research is likely to be needed to tease out the effects of particular deregulation designs on particular climate and renewable energy policy outcomes. Utility deregulation policies are enormously complex and qualitatively heterogeneous.... One possibility the authors might consider here is just excluding this particular policy from the overall policy stringency measure, because there isn’t a lot of convincing evidence that deregulation helps or hinders climate policy *overall.* The effects can be mixed and highly contingent. So I think a comprehensive stringency measure that simply excluded deregulation altogether, while incomplete, would be fully acceptable and would still offer a great improvement and contribution to the literature over current measures that focus on just one or two policies or that are “coarsely additive,” as the authors correctly argue.*

The reviewer’s comment echoes other reviewers’ (and our own) concern about researcher degrees of freedom in deciding which policies to include and how to code them. We agree with the reviewer that it is quite difficult to come up with a quantitative coding scheme for electricity market deregulation, due to the complexities that the reviewer notes. We considered an ordinal coding scheme but were unable to devise a satisfactory one. Instead we settled back to the more coarse (but also more reliable) dichotomous scheme where 1 indicates any degree of consumer choice in electricity markets, and 0 indicates that the utilities fully control electricity production, transmission, and delivery. We also note that this dichotomous coding scheme is consistent with the coding used by other authors who

have examined the influence of deregulation on renewables deployment and CO₂ emissions (Martin and Saikawa 2017, Menz and Vachon 2006, Prasad and Munch 2012).

As a robustness check in the spirit of the suggestion the Reviewer makes here, we have estimated a series of versions of the climate policy index. Each version excludes one of the policies that are included in our full index. The models were estimated in exactly the same way as the full index, except that one policy was left out each time. We then estimated the effect of climate policy on CO₂ emissions from the electricity sector with each index, as a robustness check to ensure that the results are not overly sensitive to the inclusion or exclusion of any of the policies. The resulting set of coefficient estimates (Figure 4, below), shows that our results are robust to the specific policies included in the model. The point estimates vary slightly, but they are all statistically significant ($p \leq 0.05$) and similar in magnitude. We have also added this figure and an explanation of the procedure underlying it to the Supplemental Information.

Figure 4: **Sensitivity of results to the exclusion of specific policies:** We estimated our climate policy index in a series of models, each of which excludes one policy that is included in the index used in our primary results. The figure shows the results of our main regression specification, with each version of the model used as the primary independent variable and accounting for measurement error in the index. The Y axis shows the policy that was left out of the index for each set of results.

Of particular relevance for the reviewer’s question about utility deregulation, our results do not appear sensitive to the inclusion or exclusion of utility deregulation. This is consistent with the results shown in Figure 1 in this memo, which shows that utility deregulation is not heavily weighted in the model. While we acknowledge that there is no perfect way to code deregulation quantitatively, we hope that the robustness of our results to the inclusion

or exclusion of this variable alleviates the reviewer’s concern.

3.3.3 Renewable Portfolio Standard

Though we know a lot more in the literature about these policies than we do about deregulation/restructuring, once again, the authors’ current conceptualization of RPS policy design/stringency is overly simplistic. Missing are key design elements such as resource eligibility (see Glenna and Thomas 2010), geographic restrictions and accounting rules (see Rountree 2019) and unbundled/tradable versus bundled RECs (see Yin and Powers 2010, which the authors, to their credit, do cite). In general, I recommend Fischlein and Smith (2013) for a more complete treatment of RPS policy design elements and their effects.

We appreciate the reviewer’s point that RPS policies vary on many dimensions other than whether they are voluntary or mandatory and the level of the standard, both of which we have included as inputs to the climate policy index. The Fischlein and Smith (2013) and Rountree (2019) papers to which the Reviewer refers provide excellent, detailed analyses of the various design details inherent to an RPS policy. We have included them as citations for our point in the introduction that the RPS literature has emphasized the importance of design details.

The impulse behind our model is similar to that of Fischlein and Smith (2013) and the other papers that have incorporated the details of policy design into their analyses of RPS. These papers are, at their core, attempting to measure RPS in a more nuanced way than simpler measures allow. Our model simply takes a different approach to measuring nuance. Instead of digging extensively into the details of each policy indicator (as in Fischlein and Smith (2013)) or closely following the details of a single state over time (as in Rountree (2019)), we pool information across policy indicators, over time, and across states. This choice reflects our distinctive approach to the breadth-depth tradeoff that all research in this area faces (though scholars do not always explicitly acknowledge this tradeoff).

We estimate a measure which is expansive in its breadth—it covers a long time period, all 50 states plus the District of Columbia, and reflects a wide array of policies—though it necessarily sacrifices some detail (ie, depth) at the individual policy level. In contrast, the Fischlein and Smith (2013) study is cross-sectional, and updating these authors’ dataset over time would be infeasible due to the spotty availability of state legislative text over time. Likewise, the Rountree (2019) study covers an extended time period, but only for a single state and at a level of detail that would be impossible to achieve for all states, over time, and across policies. This infeasibility is a symptom of the breadth-depth tradeoff that we faced in gathering our data and estimating this model. Instead of using a cross-sectional, extremely deep dataset for each policy, or a time-series analysis of a single policy in a single state, we use a time-series of coarser data that is available for all states. This relatively coarse (compared with Fischlein and Smith (2013) and Rountree (2019)) nationally comprehensive, cross-policy time series is feasible to obtain and maintain. Crucially, our validation checks (Supplementary Figures S2-S4) indicate that the model we estimate with this dataset is valid.

We also note that this breadth-depth tradeoff is similar to the classic tradeoff between parsimony and goodness of fit across many types of statistical models.

3.3.4 Deregulation

On p. 4, the authors mention “electricity market restructuring (ie, decoupling)” but these are NOT the same thing. Decoupling is an energy efficiency policy that divorces electricity rates from volume of sales whereas restructuring is what the authors seem to refer to in this manuscript as “deregulation.”

Thanks to the reviewer for pointing out this error in the manuscript. We have corrected the mistake in the revision. This sentence now reads, “Many states have also adopted solar tax credits, renewable portfolio standards, electric and gas system decoupling, and requirements that power plants register and record their emissions.”

3.3.5 Miscellaneous concerns

I have some limited concerns about some of the other policies included in Table 1, too, but these may be alleviated if the authors were to provide source material for all of this information.

We hope that the source material provided alleviates the reviewer’s concerns.

3.4 Time period

It would be good for the authors to explain why they chose the time period they did (2000-2020).

Thanks to the reviewer for pointing out that we had not explained this anywhere in the manuscript. We have added footnote 2 on page 2, to explain the closely related substantive and empirical reasons for our choice of time period. States began in earnest to pass climate policies in 2000, and the data are too thin prior to 2000 to reliably estimate the policy model.

3.5 Political economic considerations

On p. 2, the authors discuss the “speed and extent of the technological transformation,” but there are also economic transformations that are relevant, too, and a lot also depends on regulatory discretion. The political economy of each state is critical.

Thanks to the reviewer for highlighting that this statement falls short in encapsulating the idea we intended to convey. Here, we sought to articulate that different policies can be expected to spur different degrees of change. We did not intend to highlight the technological component of climate policy above and beyond political, economic, bureaucratic, or other considerations that shape climate policy choice and moderate policy impact. To avoid confusion without getting into an extended discussion of these myriad considerations—which are largely aside to the point we needed to make here—we have modified this introductory

sentence. The sentence no longer refers to technological shifts specifically but instead to changes in general.

3.6 References Mentioned:

- *Fischlein and Smith (2013)*. “Revisiting Renewable Portfolio Standard Effectiveness: Policy Design and Outcome Specification Matter.” *Policy Sciences* 46(3):277-310.
- *Glenna and Thomas (2010)*. “From Renewable to Alternative: Waste Coal, the Pennsylvania Alternative Portfolio Standard, and Public Legitimacy.” *Society and Natural Resources* 23(9):856-871.
- *Rountree (2019)*. “Nevada’s Experience with the Renewable Portfolio Standard.” *Energy Policy* 129:279-291.
- *Yin and Powers (2010)*. “Do State Renewable Portfolio Standards Promote In-State Renewable Generation?” *Energy Policy* 38(2):1140-1149.

We thank the reviewer for providing these references and have incorporated them into our discussion of the literature on renewable portfolio standards where appropriate. As the reviewer notes, we had already included references to Yin and Powers (2010). In the revised submission we have added all three of the other papers to our list of citations to papers focused on RPS, and we have added Rountree (2019) and Fischlein and Smith (2013) to our citations of papers that highlight the need to incorporate design differences into assessments of RPS impact.

References

- Anguelov, Nikolay, and William F. Dooley. 2019. “Renewable Portfolio Standards and Policy Stringency: An Assessment of Implementation and Outcomes.” *Review of Policy Research* 36(2): 195–216.
- Barbose, Galen, Ryan Wiser, Jenny Heeter, Trieu Mai, Lori Bird, Mark Bolinger, Alberta Carpenter, Garvin Heath, David Keyser, Jordan Macknick, Andrew Mills, and Dev Millstein. 2016. “A retrospective analysis of benefits and impacts of U.S. renewable portfolio standards.” *Energy Policy* 96(September): 645–660.
- Bowen, William M., Sunjoo Park, and Joel A. Elvery. 2013. “Empirical Estimates of the Influence of Renewable Energy Portfolio Standards on the Green Economies of States.” *Economic Development Quarterly* 27(November): 338–351.
- Carley, Sanya. 2009. “State renewable energy electricity policies: An empirical evaluation of effectiveness.” *Energy Policy* 37(August): 3071–3081.
- Carley, Sanya, Lincoln L. Davies, David B. Spence, and Nikolaos Zirogiannis. 2018. “Empirical evaluation of the stringency and design of renewable portfolio standards.” *Nature Energy* 3(September): 754–763.

- Caughey, Devin, and Christopher Warshaw. 2022. “Dynamic Democracy.” In *Dynamic Democracy*. University of Chicago Press.
- Fischlein, Miriam, and Timothy M Smith. 2013. “Revisiting renewable portfolio standard effectiveness: policy design and outcome specification matter.” *Policy Sciences* 46(3): 277–310.
- Martin, Geoff, and Eri Saikawa. 2017. “Effectiveness of state climate and energy policies in reducing power-sector CO₂ emissions.” *Nature Climate Change* 7(December): 912–919.
- Matisoff, Daniel C, and Jason Edwards. 2014. “Kindred spirits or intergovernmental competition? The innovation and diffusion of energy policies in the American states (1990–2008).” *Environmental Politics* 23(5): 795–817.
- Menz, Fredric C., and Stephan Vachon. 2006. “The effectiveness of different policy regimes for promoting wind power: Experiences from the states.” *Energy Policy* 34(September): 1786–1796.
- NC Clean Energy Technology Center. 2022. “Database of State Incentives for Renewables & Efficiency.” .
- Pastor, Daniel J. 2020. “The effects of renewables portfolio standards on renewable energy generation.” *Economics Bulletin* 40(3): 2121–2133.
- Prasad, Monica, and Steven Munch. 2012. “State-level renewable electricity policies and reductions in carbon emissions.” *Energy Policy* 45(June): 237–242.
- Rountree, Valerie. 2019. “Nevada’s experience with the Renewable Portfolio Standard.” *Energy policy* 129: 279–291.
- Shrimali, Gireesh, Melissa Lynes, and Joe Indvik. 2015. “Wind energy deployment in the U.S.: An empirical analysis of the role of federal and state policies.” *Renewable and Sustainable Energy Reviews* 43(March): 796–806.
- Yi, Hongtao. 2015. “Clean-energy policies and electricity sector carbon emissions in the U.S. states.” *Utilities Policy* 34(June): 19–29.
- Yi, Hongtao, and Richard C Feiock. 2014. “Renewable energy politics: policy typologies, policy tools, and state deployment of renewables.” *Policy Studies Journal* 42(3): 391–415.
- Yin, Haitao, and Nicholas Powers. 2010. “Do state renewable portfolio standards promote in-state renewable generation?” *Energy Policy* 38(February): 1140–1149.
- Zhou, Shan, and Barry D. Solomon. 2020. “Do renewable portfolio standards in the United States stunt renewable electricity development beyond mandatory targets?” *Energy Policy* 140(May): 111377.

REVIEWER COMMENTS

Reviewer #3 (Remarks to the Author):

At the outset, I would like to say that I appreciate the authors' thoughtful responses to my concerns. Indeed, the format and approach they took in their response letter is excellent, and extremely thorough, and I plan to emulate it in my own responses to reviewers on future manuscripts.

Next, I am satisfied with the sources provided for their measures for the 26 policies, with the exception of the websites cited for the "utility deregulation policy," which, as someone with significant expertise in this area, I had never heard of, and, upon looking at them, I suspect that they are industry-funded and selective in their understandings of the implications of this sweeping policy regime distinction.

If the authors' definition is indeed, as they write in their memo, "1 indicates any degree of consumer choice in electricity markets, and 0 indicates that the utilities fully control electricity production, transmission, and delivery," then I must reject their coding on this measure, since TX offers the most competitive electricity system in the country (which is partly why that state experienced the blackouts it did with Winter Storm Uri, by I digress) and CA also has retail choice (<https://www.cpuc.ca.gov/consumer-support/consumer-programs-and-services/electrical-energy-and-energy-efficiency/community-choice-aggregation-and-direct-access->). Both states have had this in place for well over a decade.

If the authors have an interest in educating themselves further about the complexities involved in utility deregulation, better termed restructuring, then I would point them to Isser (2015, Cambridge University Press), *Electricity Restructuring in the United States*. To be clear, I'm not asking the authors to read this book, nor am I asking them to cite it in a revision; I'm simply providing it for their reference, and to help illustrate why this doesn't lend itself to a dichotomous variable.

All of that said, I do appreciate the authors' additional robustness checks. It is clear that this variable isn't doing much of the work in their model anyway (from the authors' response letter, p. 15: "utility deregulation is not heavily weighted in the model."). Therefore, if the authors' goal is prompt publication in *Nature* of what is overall an excellent contribution, then I'd recommend simply omitting this from their 26 policies (they'd be left with 25—still very impressive!), assuming the regression results hold. If they'd like to say something about why they are omitting it, they may.

In the authors' response letter, I also appreciate the authors' framing of, as they put it so well, "the breadth-depth tradeoff." This is also the spirit of their new footnote 4.

Clearly, in this case, the authors are opting for breadth at the expense of depth. That is fine. For an analysis that *does* necessarily sacrifice depth for breadth, theirs is exceptionally well-done. The breadth is extremely impressive. The breadth alone makes it worthy of publication, in my opinion—even as I'm one whose own work likely sacrifices breadth for depth, which admittedly has shortcomings of its own. Both are sins that are sometimes necessary to further overall knowledge in the field, in my view, as I would hope the authors would agree.

However, what I would ask of the authors is just that they be more explicit about this tradeoff—not just in a confidential letter to reviewers, but in the manuscript itself. And not in just a footnote either. This should really go in a more prominent place, such as the Discussion section. I think doing this may actually alleviate some of my fellow reviewers' concerns as well, so it is likely the "path of least resistance" to prompt publication, and I would very much like to see this published so that I can cite it in my own work.

In this spirit, it would also be helpful to state explicitly what is being omitted for the sake of

parsimony. This language is already available in the authors' response memo, in particular in their response 2.1 to Reviewer #2. The authors write here, and in the manuscript itself but only very briefly and without elaboration or discussion of tradeoffs, on p. 6, they are limited to "dimensions across which states are comparable." This is a perfectly justifiable analytic strategy, but I worry that the way the manuscript currently reads will dissuade researchers interested in studying particular states in ways that are **not** comparable across all fifty from doing so (see, as just one example, the Rountree 2019 article). It may also dissuade those who study just a few states, but in great depth (Stokes 2020 book comes to mind), from citing the present piece to justify, for instance, their case selection.

Surely, we wouldn't want to do that, because then we'd never learn about the unique political conditions that may allow a state like California to go so far beyond the others (or to understand why a laggard state may be such a laggard, other than easy-to-reach explanations such as partisanship).

So, here again, my wish is for the authors to acknowledge what they sacrifice for the sake of parsimony and breadth, understanding that it is a worthy sacrifice, given the power of what they **are** doing, which they do a nice job of stating.

Finally, in the spirit of maximizing the usefulness of this to future researchers, could the authors include a table that compares where each of the fifty states falls numerically on their overall holistic index, in the years 2000, 2010 (or maybe, space-permitting, 2007 and 2014, which are used in Figure 2, instead of 2010), and 2020? Figure 1 is difficult to read because of all of the lines, and not all fifty states are clearly labeled on the axis even though they each have a line. Figure 2 is in the spirit of what I'm asking for, but it is difficult to see precisely where each state is in relation to the others, so I would appreciate a table (perhaps in the appendix?) in addition to Figure 2.

Memo for the Reviewers of “US State Climate Policy Commitments, the Energy System, and the Economy”

We appreciate the opportunity to revise and resubmit our manuscript and are grateful to the reviewer for their sustained engagement with our study. In response to the reviewer’s helpful comments, we have made the following changes to the manuscript:

- As Reviewer 3 noted, deregulation is quite complex. The reviewer suggested removing it rather than reducing it to a dichotomous variable for inclusion in our index. Since our robustness checks from the prior round of revisions showed that this policy was not pivotal to the index, we have removed it in the current submission. The results are robust to the removal of this policy from our index. All figures and tables have been updated to reflect this change.
- Reviewer 3 reiterated the concern that we had inadequately discussed our decision to focus on breadth in our policy measurement strategy, at the expense of sacrificing depth. We have addressed this concern in two ways. First, we have added language acknowledging this tradeoff to the Introduction. We have moved the content of Footnote 4 from the previous draft into the introduction section, as suggested by the editor. We have also added to the introduction some of the language from Sec. 2.1 of our initial response to reviewers, as suggested by Reviewer 3. The penultimate paragraph of the introduction now reads:

In gathering and coding our data, we strove to strike a balance between breadth (ie, comprehensiveness over time and across states) and depth (ie, faithful depictions of finely grained distinctions between specific states’ policy designs). Our granular coding scheme cannot capture all distinctions between states’ versions of different policies, nor does it exhaustively quantify everything that every state is doing to address climate change. Instead it provides a holistic ranking of state climate policy efforts on the dimensions across which states are comparable. The benefit of the index is that, by pooling information across many measures, we produce a set of estimates that is more comprehensive than any single-policy measure would be.

Second, in the discussion, we highlight opportunities for future research focused on political dynamics that are particular to single policies contained in our index. The discussion now includes this pair of sentences:

In this spirit, scholars should also continue to probe deeply the political dynamics that are particular to specific policies contained in our index. Such work provides a crucial complement to our broad approach to policy measurement, which necessarily obscures deep analysis of particular policies including those that are only adopted in a handful of states.

We hope that this inclusion will prevent our paper from deterring scholars who are interested deep analysis of a small number of states or a small number of policies.

- Reviewer 3 noted that a detailed table of descriptive results would make our data more accessible for interested readers. In response to the helpful suggestion to provide detailed descriptive results in tabular form, we have added Supplementary Table 1. This table shows our estimate of each state’s climate policy stringency in 2000, 2010, and 2020, along with each state’s ranking on our stringency measure, compared with the other states in the same year. We have noted this addition in a footnote on Page 7, where we introduce descriptive Figure 1.

REVIEWERS' COMMENTS

Reviewer #3 (Remarks to the Author):

Thank you to the authors for making the suggested changes. I am happy to recommend that this manuscript now be accepted for publication in Nature Communications. This is a very important contribution to the literature on U.S. state-level climate policy.